# Gross, Histologic and Immunohistochemical Characteristics of Keratoacanthomas in Lizards

**DOI:** 10.3390/ani13030398

**Published:** 2023-01-24

**Authors:** Ferran Solanes, Koen Chiers, Marja J. L. Kik, Tom Hellebuyck

**Affiliations:** 1Department of Pathobiology, Pharmacology and Zoological Medicine, Faculty of Veterinary Medicine, Ghent University, Salisburylaan 133, B-9820 Merelbeke, Belgium; 2Department of Biomedical Health Sciences, Pathology Division, Pathology Exotic Animals and Wildlife, Faculty of Veterinary Medicine, Utrecht University, Yalelaan 1, 3584 CL Utrecht, The Netherlands

**Keywords:** cutaneous, immunohistochemistry, keratoacanthoma, lizards, neoplasia, squamous cell carcinoma

## Abstract

**Simple Summary:**

Tumors of the skin are one of the most commonly observed neoplasms in captive lizards. The current study characterizes keratoacanthoma, a previously undescribed skin tumor, in five male lizards (one bearded dragon, one veiled chameleon, and three panther chameleons) with an average to high age. In all lizards, keratoacanthomas presented as cystic nodules with a central keratin pearl that was predominantly located at the body wall. In all chameleons, a multicentric distribution was observed. Following surgical removal of the keratoacanthomas in all lizards, a follow-up period of one to two years was established. While the skin neoplasia reappeared in the bearded dragon and the veiled chameleon, no recurrence was seen in the panther chameleons. Keratoacanthoma constitutes a rather benign histologic variant of squamous cell carcinoma, representing a non-invasive but rapidly growing skin neoplasia that may be associated with the inappropriate use of ultraviolet lighting in the captive environment. In addition, panther chameleons may show a species predisposition as well as a tendency to develop multicentric keratoacanthomas. The present study delivers pertinent results for the diagnosis, prevention, and treatment of keratoacanthomas in lizards.

**Abstract:**

The present study describes the clinical behavior as well as the histopathologic and immunohistochemical characteristics of keratoacanthomas (Kas) in three different saurian species. While Kas presented as two dermal lesions in a bearded dragon (*Pogona vitticeps*), multicentric Kas were observed in three panther chameleons (*Furcifer pardalis*) and a veiled chameleon (*Chamaeleo calyptratus*). Macroscopically, Kas presented as dome-shaped skin tumors with a centralized keratinous pearl and a diameter ranging from 0.1–1.5 cm. In all lizards, Kas were predominantly located at the dorsolateral body wall, and KA of the eyelid was additionally observed in three out of four chameleons. Histologically, KAs presented as relatively well-defined, circumscribed epidermal proliferations that consisted of a crateriform lesion containing a central keratinous pearl with minimally infiltrating borders. In all KAs, a consistent immunohistochemical pattern was observed, with the expression of cyclooxygenase-2, E-cadherin, and pan-cytokeratin. A follow-up period of one to two years was established in all lizards. While no recurrence was observed in the panther chameleons, recurrence of a single keratoacanthoma was observed in the bearded dragon after one year, and in the veiled chameleon, multicentric keratoacanthomas reappeared during a follow-up period of two years. We describe KA as a previously unrecognized neoplastic entity in lizards that constitutes a low-grade, non-invasive but rapidly growing skin tumor that may show a multicentric appearance, especially in chameleons. As previously postulated for dermal squamous cell carcinomas (SCC), artificial ultraviolet lighting may play an important role in the oncogenesis of KAs in lizards. Although dermal SCCs in lizards show similar predilection sites and gross pathologic features, our results suggest that KA should be considered as a histologic variant of SCC that represents a rather benign squamous proliferation in comparison to conventional SCCs. Early diagnosis of KA and reliable discrimination from SCCs are essential for the prognosis of this neoplastic entity in lizards.

## 1. Introduction

Although once considered to be uncommon, neoplasia in reptiles is routinely encountered in veterinary practice, with the hematopoietic, hepatobiliary, and integumentary systems being most frequently affected [1,2]. Chromatophoromas and squamous cell carcinomas (SCCs) represent the predominant skin neoplasia in captive squamates [3,4], with SCC showing a presumptive species predisposition in bearded dragons (*Pogona vitticeps*) and panther chameleons (*Furcifer pardalis*) [2,5,6,7]. Keratoacanthoma (KA) has been described in humans, dogs, and birds, especially in broiler chickens, as a well-differentiated histologic variant of SCC [8,9,10,11]. In dogs, KA is either referred to as infundibular keratinizing acanthoma (IKA) or subungual KA depending on its localization [10,11,12]. Although correct histologic characterization allows differentiation of KA from SCC, some controversy remains concerning the correct classification of KAs, as some consider it to be a precancerous stage of dermal SCC or a pseudo-cancerous lesion [13,14,15,16]. The present study describes the clinical, histologic, and immunohistochemical (IHC) characteristics of KAs in lizards.

## 2. Materials and Methods

### 2.1. Animals

Five unrelated lizards that were part of captive collections were presented at a veterinary teaching hospital between 2020 and 2022 because of showing nodular skin lesions. The lizards included one bearded dragon, one veiled chameleon (*Chamaeleo calyptratus*), and three panther chameleons (Table 1). In all cases, tissue samples were collected from the skin nodules following in toto excision after intravenous (IV, jugular vein) induction of anesthesia with 10 mg/kg alfaxalone (Alfaxan^®^ Multidose, 10 mg/mL, Jurox Limited, Crawley, UK). Anesthesia was maintained with 1.5–2.0% isoflurane (Isoflo^®^, Abbott Logistics B.V., Breda, The Netherlands) in 1 L medical oxygen with intermittent positive-pressure ventilation. All dermal nodules were surgically removed with resection margins of, on average, 1 mm in small lesions and 3 mm in the larger lesions. Routine closure of the skin was performed using a simple everting pattern with a 5-0 absorbable suture (Monocryl^®^, Ethicon, Raritan, NJ, USA). All samples were fixed in 10% neutral buffered formalin for 24–36 hours for histopathological evaluation and IHC staining.

### 2.2. Histopathology

Following dehydration and embedding of tissues into paraffin blocks, 5-µm-thick sections were cut and stained with hematoxylin and eosin (HE). All histological sections were confirmed as neoplastic and further characterized histologically. Mitotic figures were counted in 10 high-power fields (HPF) in randomly selected areas, and the mean numbers were calculated. The degree of nuclear atypia was categorized as mild, moderate, or marked if less than 30%, between 30–60%, or more than 60% of the neoplastic cells showed nuclear atypia, respectively.

### 2.3. Immunohistochemistry

IHC staining for cyclooxygenase-2 (COX-2), E-cadherin, and pan-cytokeratin (Pan-CK) in all eyelid lesions and one body wall lesion per lizard was performed. Paraffin-embedded dermal tissue blocks were cut into 5 μm sections and mounted on 3-aminopropyltriethoxysilane-coated slides. Next, slides were deparaffinized and rehydrated in xylene and decreasing concentrations of alcohol in H_2_O (100, 96, 50, and 100% H_2_O, respectively). 

Antigen retrieval was performed by immersion in citrate-buffered (0.01 M, pH 6) distilled water and microwaving for 3.5 min at 850 W and 10 min at 450 W. Next, slides were allowed to cool down for 20min and incubated with H_2_O_2_ (S202386-2; Agilent, Santa Clara, CA, USA) at room temperature for 5 min. Subsequently, incubation with the primary monoclonal mouse COX-2 (1/20, 610204, BD Biosciences, Franklin Lakes, NJ, USA)/E-cadherin (1/100, M3612, Agilent)/pan-CK (1/50, M3515, Agilent) antibodies (1:200; ab7778; Abcam, Cambridge, UK) was performed at room temperature for 30 min with background-reducing components (S302283-2; Agilent). Followed by incubation with a polymer-based anti-mouse secondary antibody (K400111, Agilent) at room temperature for 30 min, visualization was performed in a 3.3-diaminobenzidine solution (K346811, Agilent) at room temperature for 5 min. The cell nuclei were counterstained with hematoxylin, rinsed in tap water and dehydrated and coverslips were applied. In between all steps, the sections were washed extensively and repeatedly with phosphate-buffered saline.

Negative controls consisted of omitting the primary antibody in normal skin samples from a dog and a bearded dragon. 

To evaluate the expression of COX-2, E-cadherin, and cytokeratin, an immunoreactive score system (IRS), based on the percentage of positive cells and intensity of staining according to Fedchenko et al., was used (Table 2) [17].

## 3. Results

### 3.1. Clinical History and Gross Pathology

All examined lizards were males with an age ranging from 3–6 years. In general, the captive management of all cases was deemed adequate. In all lizards, however, the minimum safe distance to the ultraviolet (UV-B) source was considered inappropriate, taking into account manufacturer recommendations based on the natural UV-B exposure values that have been described for the involved species [18]. Based on the clinical history, first detection of dermal lesions by the owner ranged from two months to two years prior to initial presentation (Table 1). In all cases, the lesions initially presented as skin nodules with a smooth surface and greyish discoloration that rapidly developed into whitish-greyish crateriform nodules with a keratinous core. In most cases, the owners did not seek veterinary advice until multiple dermal nodules were noticed, or until one or more dermal nodules reached a considerable size.

While 18, 15, and 3 lesions were noted in panther chameleon 1, 2, and 3, respectively, the veiled chameleon was presented with four dermal nodules, and two dermal nodules were detected in the bearded dragon (Table 1). The number of dermal lesions seemed to be positively correlated with the interval between first detection of the lesions and initial presentation. Besides panther chameleon 1, which showed a poor body condition and sunken eyes, indicating more than 8–10% of dehydration [19], none of the remaining lizards displayed other clinical signs in addition to the skin lesions.

While the dermal lesions were located at the dorsolateral body wall in all lizards (Figure 1A–D), a single, unilateral nodular lesion of the eyelid was additionally present in the veiled chameleon, panther chameleon 1, and panther chameleon 3 (Figure 2). The diameter of the lesions ranged from 0.1–1.5 cm, with an average diameter of 0.33 cm. In the bearded dragon and panther chameleon 3, the average diameter of the lesions was 0.56 cm. In the chameleons with multicentric distribution, on average two large lesions with an average diameter of 1 cm (Figure 1A and Figure 2) were present and the remaining lesions consisted of small nodular lesions with a diameter of 0.1–0.2 cm (Figure 1B,C). 

During surgical removal, it was noted that all dermal nodules were well-demarcated and did not infiltrate the subcutis. Especially in the smaller nodules, the central keratin plug detached spontaneously when pressure was performed during surgical removal (Figure 1B).

Recovery from anesthesia was uneventful in all cases and a follow-up period of one to two years was established in all lizards (Table 1). When the lizards returned to their owners, management advice was provided with an emphasis on the optimization of the provision of UV-B lighting according to the recommended values. While the owners of panther chameleons followed these recommendations, the owners of the veiled chameleon and the bearded dragon did not make adjustments. In the bearded dragon, a new dermal nodule developed in approximately the same location as the first tumor after 1 year, and in the veiled chameleon, multicentric skin nodules located at the lateral body wall and the head, including both eyelids, reappeared during a 2-year follow-up period (Figure 3). As the nodules interfered with the normal feeding behavior, the chameleon deteriorated and was euthanized by a local veterinarian.

### 3.2. Histopathology and IHC

Histologic examination of eleven skin lesions revealed a characteristic architectural pattern in all five cases. Nodular lesions presented as a relatively well-defined, circumscribed epidermal proliferation including a multilobular, exo-endophytic cyst-like invagination of the epidermis that creates a crateriform lesion with a central keratinous plug with minimally infiltrating borders (Figure 4). Peripheral to the keratin-filled crater, lip-like borders of well-differentiated squamous cells with a low degree of pleomorphism were observed (Figure 5). Areas of pseudoepitheliomatous hyperplasia that formed folds inside the crater and the adjacent dermis were noticed. Depending on the section, those folds presented as buds, cords, or isolated islands, centered by orthokeratotic pearls. Based on these findings, a final histological diagnosis of KA was made for all examined skin nodules that were obtained from the lizards. 

Neoplastic cells were typically enlarged and showed a pale eosinophilic ground glass-like cytoplasm (Figure 5). All examined KAs showed a mitotic index ranging from 0–2 and absent to mild nuclear atypia. At the base of 80% of the KAs, a mild inflammatory infiltrate with heterophiles, lymphocytes, and macrophages, as well as small keratin pearls, was present.

Histological sections of the eyelid nodule in panther chameleon 1 showed two components. While the central area of the process clearly showed KA features with a mitotic index of 1, low cellular atypia, and pleomorphism (Figure 6A), the characteristics of the neoplastic cells at the periphery were fully compatible with well-differentiated SCC, including a mitotic index of 4, marked nuclear atypia, and pleomorphism (Figure 6B). A final histopathological diagnosis of KA with malignant transformation was made, similar to what has been previously described in humans by Sánchez Yus et al. [20] and Weedon et al. [21]

The results of the IHC study are presented in Table 3. Strong staining with E-cadherin (IRS score 9–12) was noted along the plasma membrane of the six examined Kas, indicating high integrity of cell-cell adhesion (Figure 7A). Moreover, strong positivity for Pan-CK (IRS 9–12) (Figure 7B) and moderate cytoplasmic immunoexpression for COX-2 (IRS 4–8) were noted (Figure 8A). In the KA with malignant transformation, strongly positive immunoexpression for COX-2 (IRS 9–12) was present in the SCC area of the process (Figure 8B) in contrast to the central part that showed KA features. 

## 4. Discussion

The present study describes the clinical behavior and macroscopic appearance as well as the histopathologic and IHC characteristics of KAs in three different saurian species. Besides KA of the spectacle in a boa (*Boa constrictor*) [22], KA has not previously been described in reptiles. Based on the findings of the present study, however, the description of the disorder of the spectacle in the boa does not seem to meet the clinical, macroscopic, and histologic criteria that are fundamental to obtaining a definitive diagnosis of KA, and the lesion rather resembled retention of multiple layers and ulceration of the spectacle associated with impaired shedding. 

It is noteworthy that all affected lizards in the present study were males with an adult to relatively old age according to physiological lifespan reference intervals [23,24,25]; this is similar to what has been reported for KA in humans and dogs [26,27,28,29]. In avian species, however, KA is almost exclusively observed in juvenile broiler chickens without an obvious gender predisposition [9]. Despite the relatively limited number of cases in the present study, male panther chameleons with an average to old age may be predisposed to the development of KA. A similar age and gender predisposition seems to exist for multicentric SCCs in panther chameleons [6].

Dermal SCC is a malignant neoplasia of epidermal keratinocytes that may exhibit various degrees of differentiation and invasiveness, as well as metastatic potential [30]. Based on this variation, the World Health Organization (WHO) Classification of skin tumors proposed by the International Agency for Research on Cancer (IARC) categorizes different histologic variants of dermal SCC with acantholytic SCC, desmoplastic SCC and spindle cell SCC representing high-risk histologic variants, and verrucous SCC and KA representing low-risk histologic variants [8,30,31]. KA is defined as a well-differentiated, minimally invasive neoplasia with low metastatic potential [9,32,33]. In contrast to dogs and birds, KA in humans often shows a spontaneous regression phase that may occur within 4–6 months of the initial diagnosis [34]. In the present study, signs of spontaneous regression were observed in none of the lizards, even in the cases that had a 6-month to 2-year interval between the first detection of skin lesions and initial presentation; in two cases, recurrence was observed. Although spontaneous regression of human KA is often observed, cases wherein KA is observed as a pre-stage of malignant SCCs [14] have also been documented [35]. If malignant transformation of KA into SCC takes place in humans, it seems to occur within a couple of months of KA development. This transformation is considered to occur spontaneously and is mainly seen in senile skin with actinic degeneration or immunocompromised persons [33]. It is unknown, however, what proportion of human KAs eventually transform into SCC and what the deciding factors for transformation are [34,36,37]. In the present study, a skin nodule of the eyelid was diagnosed as a KA with malignant transformation in panther chameleon 1, which was presented 2 years after the first detection of skin lesions (Table 1 and Figure 2). In the other cases, the interval between detection and initial presentation ranged from 2 months to 6 months, and no signs of transformation into SCC could be demonstrated. The latter findings suggest that transformation of KA into SCC does not seem to rapidly occur in the examined lizard species, and if it occurs, it can rather be expected in chronic cases. Multicentric SCC has been reported in panther chameleons, a veiled chameleon, and a bearded dragon [2,5,6]. The multicentric SCCs that were described in panther chameleons by Meyer et al. [6] may have initially comprised Kas, and the case referred to as a multicentric in situ carcinoma was most likely a KA, based on the description that is provided. As the time between the development of the primary skin lesions and the diagnosis of the multicentric SCCs was not reported, it is impossible to state whether if malignant transformation had occurred it would have initially comprised KAs in the described cases [6]. 

Based on the distribution of KAs in the lizards in the present study, mainly areas of skin that are most frequently exposed to UV-B irradiation were affected. From a behavioral point of view, chameleons typically can be classified as occasional sun baskers living in leafy shrubs [18,38]. They create a larger body surface by basking in a lateral position and flattening the coelom to a maximum in order to increase their exposure to UV-B irradiation and/or their body temperature [38]. Although artificial UV-B lighting is considered an essential part of captive management, especially for insect- and herbivorous reptiles, exposure to high-intensity focused bundles of artificial UV-B irradiation often surpasses the exposure values in the wild [6,18]. As previously postulated for dermal SCCs, BCCs, chromatophoromas, and hemangiomas, the inappropriate use of UV-B lighting sources is considered one of the most important predisposing factors towards the development of these dermal neoplasia in reptiles [3,4,6,7]. In all cases in the present study, UV-B exposure was considered inappropriate based on the recommended UV-B values for the involved species [18,38]. As the latter may play an important role in the oncogenesis of KA in lizards, we consider the provision of well-balanced UV-B irradiation essential for the prevention of this neoplastic disorder in captive lizards.

In humans, the IHC study of keratin 16 (K16) expression demonstrated that KA originates from the outer root sheath cells below the infundibulum of the hair follicle [14,39]. While reptiles do not possess hair follicles, lacunar cells are typically seen as an intermediate, undifferentiated keratinocyte cell type that appears during the epidermal renewal phase (the second phase of the physiological shedding cycle) and regeneration of the skin following traumatic insults [40]. As these lacunar cells also express K16 [41], they may also play an important role in the development of KA in lizards, especially when lacunar cells are exposed to excessive levels of UV-B irradiation and/or other traumatic insults to the skin [41,42].

All KAs in this study showed strongly positive cell membrane reactivity (IRS score 9–12) for E-cadherin, indicating high integrity cell-cell adhesion and thus low migratory potential, in contrast to what is seen in SCCs and other malignant neoplasia [31,43,44]. The strong Pan-CK immunostaining (IRS score 9–12) contributes to the delineation of KA margins and assesses the integrity of the basement membrane [45]. In contrast to the moderate COX-2 expression of KAs in the lizards in this study (Figure 7A), strong COX-2 staining was observed in the KA with malignant transformation in panther chameleon 1, similar to what has been described in humans and bird SCCs, indicating increased cell proliferation, angiogenesis, and invasiveness [33,37,43,44,46,47,48]. Based on these IHC results, COX-2 and E-cadherin staining especially can serve as an important tool in the diagnosis of KA and its differentiation from KA with malignant transformation and SCC in lizards.

If left untreated, large KAs may severely impair normal vision if located at the eyelid, and may eventually interfere with normal foraging and feeding behavior. This study demonstrates that complete excision of KA in lizards is a highly feasible and safe treatment, even in cases that show large and multicentric KAs, KAs at delicate sites such as the skin of the eyelid in chameleons, or KA with malignant transformation. Long-term follow-up was achieved in all cases of the present study (Table 1). While no recurrence was seen in the three panther chameleons, KAs reappeared in the veiled chameleon and the bearded dragon. Presumably, recurrence was associated with continued exposure to inappropriate levels of UV-B irradiation. Although longer follow-up periods are necessary, we consider the provision of optimal UV-B irradiation in captive lizards essential for the prevention of the development and recurrence of KA. 

## 5. Conclusions

This study describes the biological behavior and histopathological and IHC characteristics of KA, a previously undescribed neoplastic entity in lizards. KA seems to mainly affect adult male lizards and the dorsolateral body wall seems to be a predilection site. Chameleon species that show lateral basking behavior might make them predisposed to the development of multicentric KAs if they are exposed to inappropriate UV-B levels in the captive environment. As demonstrated in one of the panther chameleons, malignant transformation of KA can occur in a chronic stage. Even in chronic stages and cases with multicentric KAs, surgical treatment carries a good prognosis. Correct histologic characterization and IHC staining allow the differentiation of KA from conventional SCCs in lizards, although in some cases the exact classification of KA with a variable degree of malignant transformation may remain challenging; this is a similar situation to that encountered in humans.

## Figures and Tables

**Figure 1 animals-13-00398-f001:**
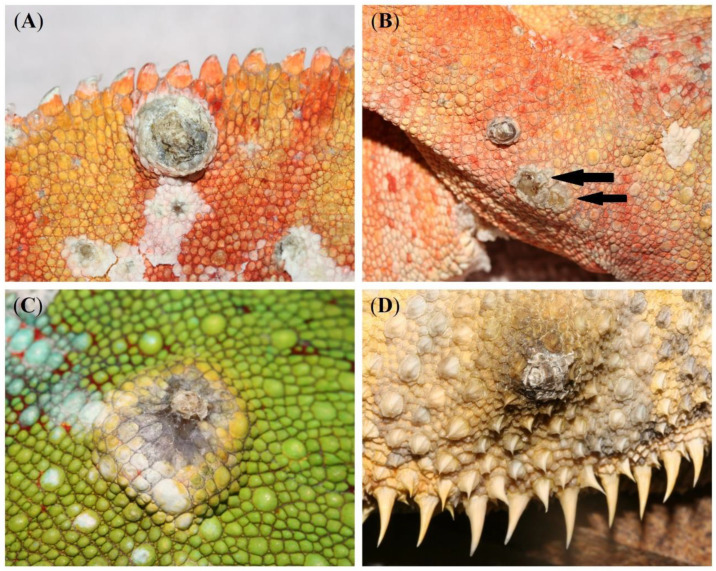
Nodular, crateriform dermal lesions located at the dorsolateral body wall in two panther chameleons (*Furcifer pardalis*) (**A**–**C**) and a bearded dragon (*Pogona vitticeps*) (**D**) that were histologically diagnosed as keratoacanthoma (KA). KAs presented as whitish-greyish crateriform nodules with a keratinous core. Especially in smaller lesions, the central keratin plug detached spontaneously when digital pressure was applied (arrows) (**B**).

**Figure 2 animals-13-00398-f002:**
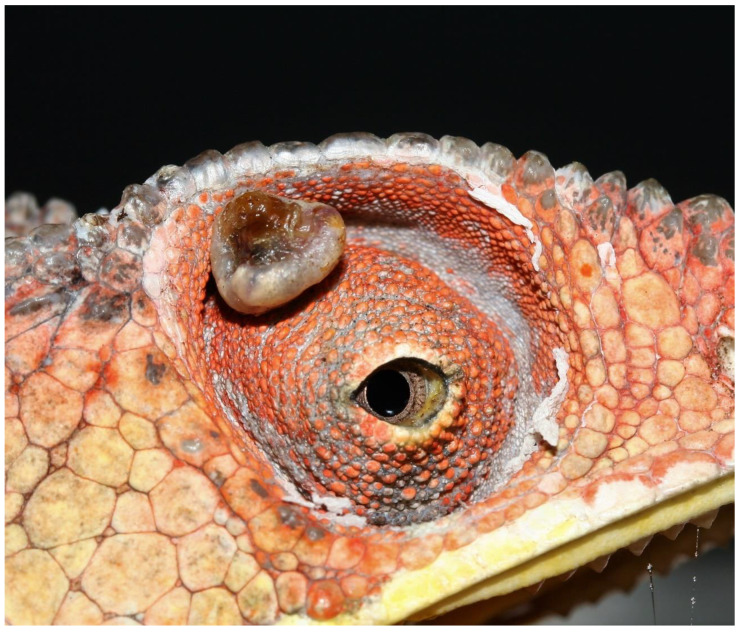
Macroscopic multicentric nodular, crateriform dermal lesion located at the dorsal part of the eyelid in panther chameleon 1 (*Furcifer pardalis*) that was histologically diagnosed as a keratoacanthoma with malignant transformation.

**Figure 3 animals-13-00398-f003:**
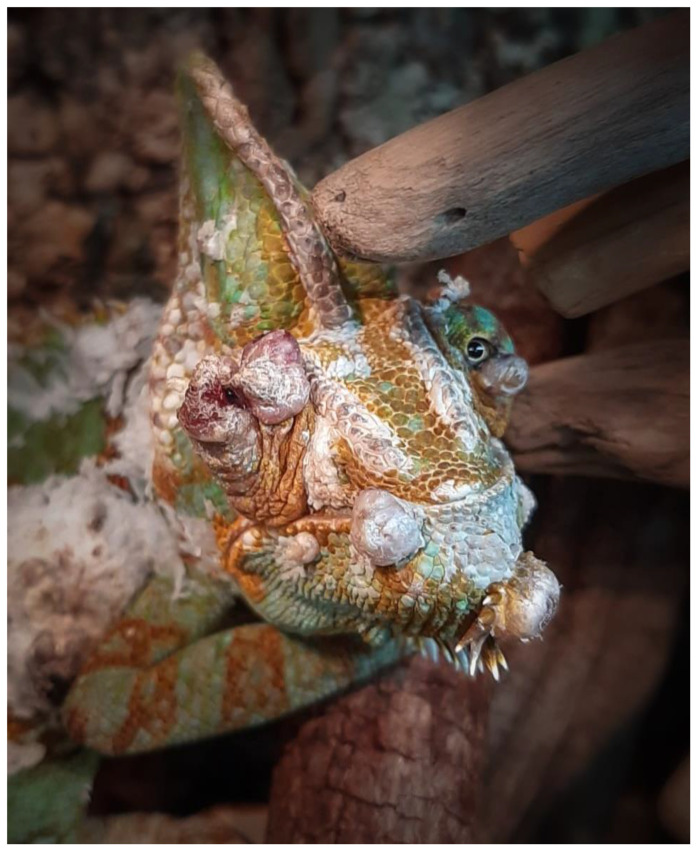
Macroscopic multicentric nodular, crateriform dermal lesions located at the head, including both eyelids, in a veiled chameleon (*Chamaeleo calyptratus*).

**Figure 4 animals-13-00398-f004:**
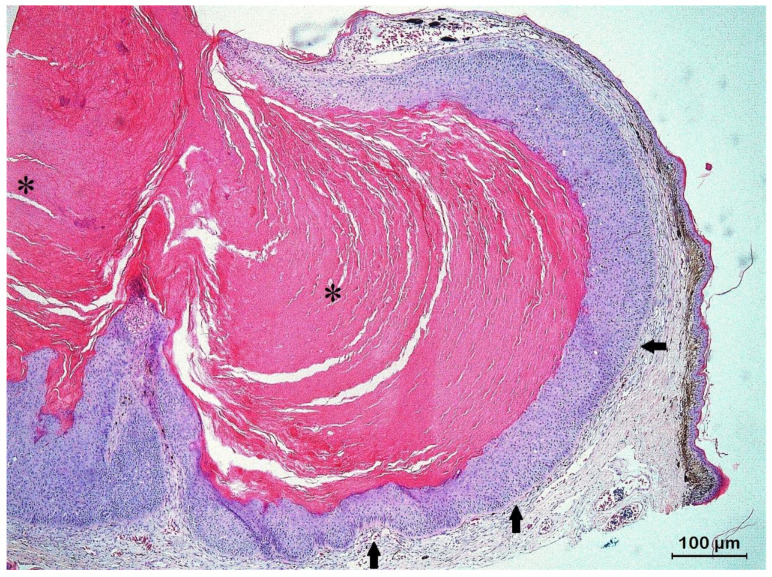
Histologic section of a keratoacanthoma (KA) from the dorsolateral body wall in a veiled chameleon (*Chamaeleo calyptratus*) showing a characteristic architectural pattern consisting of an exo-endophytic, cyst-like invagination of the epidermis that creates a crateriform lesion with a central keratinous plug (asterisks) and minimally infiltrating borders (arrows).

**Figure 5 animals-13-00398-f005:**
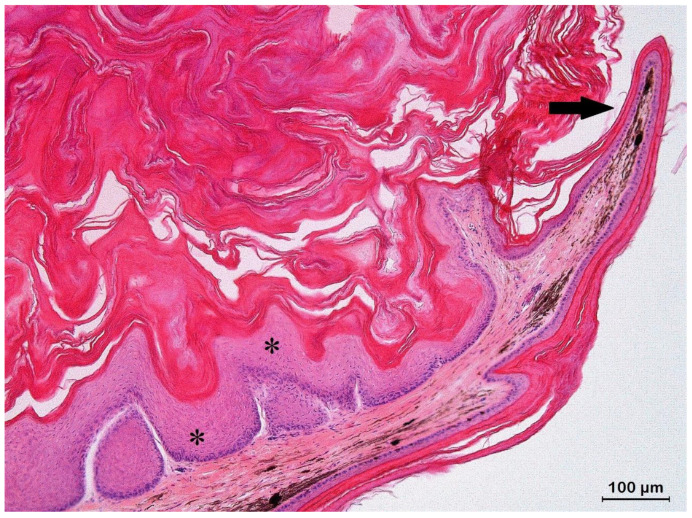
Histologic section of a keratoacanthoma in a panther chameleon (*Furcifer pardalis*) from the dorsolateral body wall. A characteristic epithelial lip (arrow) is present at the periphery, which partially extends over the central keratin plug. Note the typical large, pale pink cells with a glassy appearance (asterisks) surrounded by a thin layer of basophilic cells of the epithelial lip.

**Figure 6 animals-13-00398-f006:**
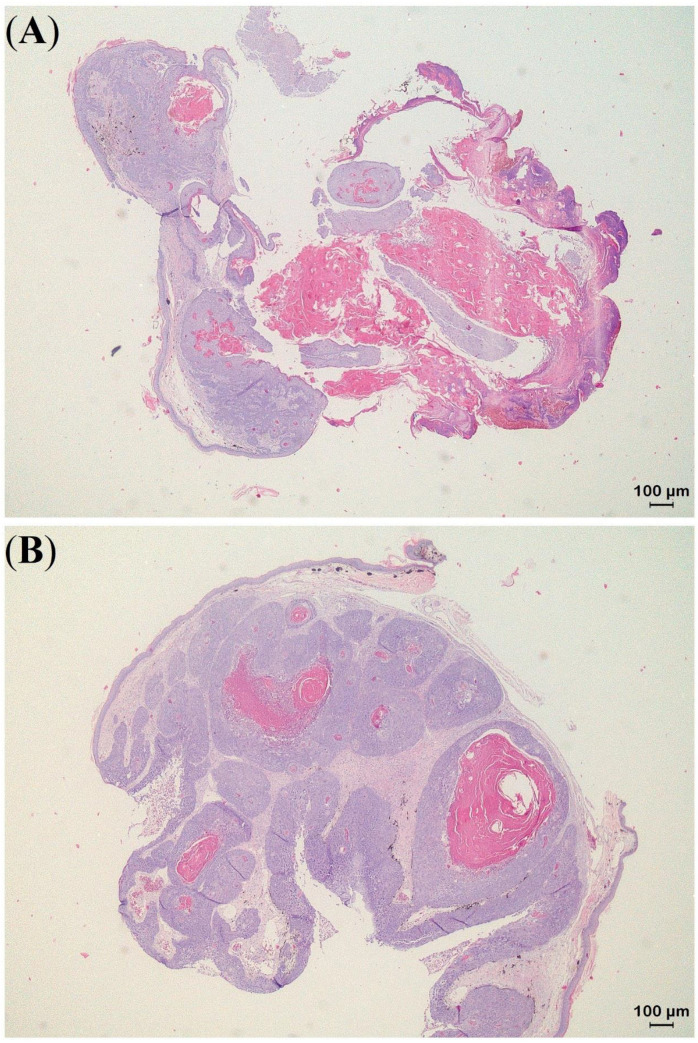
Histologic sections of a keratoacanthoma (KA) with malignant transformation obtained from the eyelid in panther chameleon 1 (*Furcifer pardalis*) with KA features in the central area (**A**) and a well-differentiated SCC at the periphery of the process (**B**).

**Figure 7 animals-13-00398-f007:**
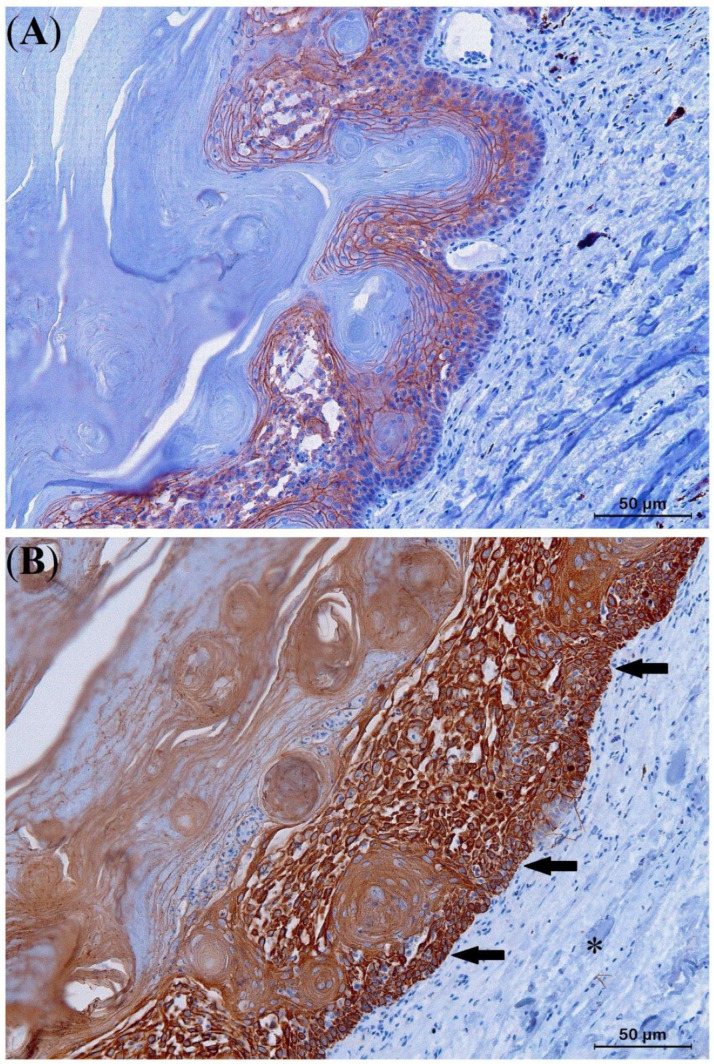
Immunohistochemical results of a keratoacanthoma (KA) from the lateral body wall of a panther chameleon (*Furcifer pardalis*) (**A**,**B**). Intense E-cadherin staining with immunoreactivity at the level of the plasma membrane of all neoplastic cells (**A**). Strongly positive pan-cytokeratin staining showing specific intracytoplasmic immunoreactivity abruptly delineating the neoplastic tissue (arrows) from the healthy skin (asterisk) (**B**).

**Figure 8 animals-13-00398-f008:**
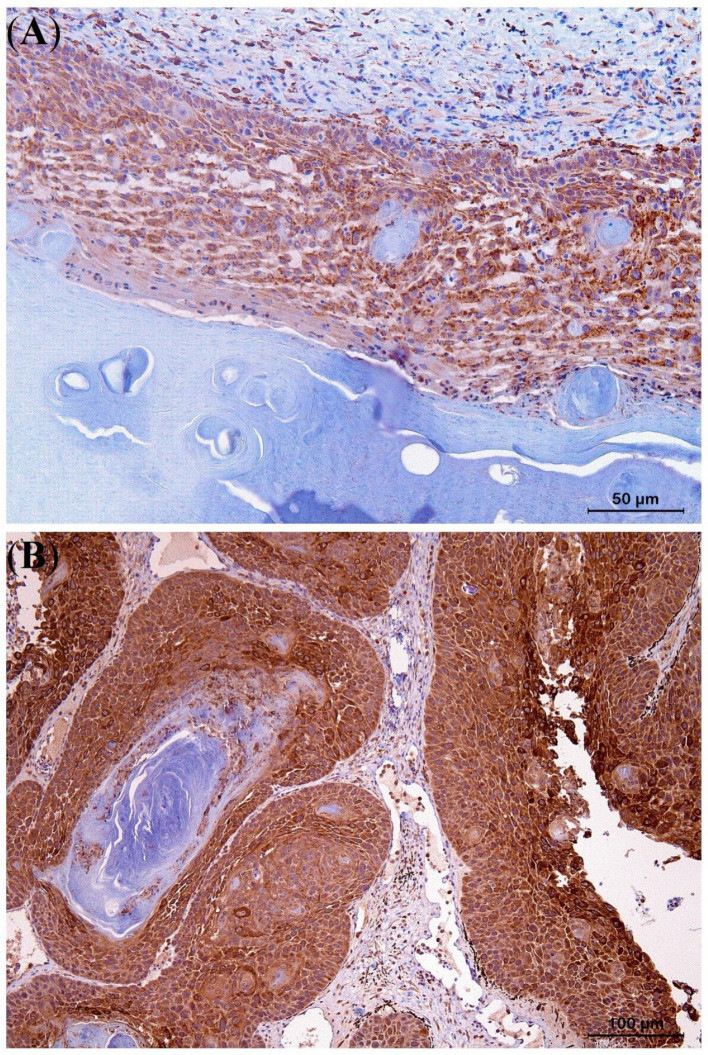
Moderate intracytoplasmic COX-2 staining of a keratoacanthoma (KA) from the lateral body wall of a panther chameleon (*Furcifer pardalis*) (**A**). Strong intracytoplasmic COX-2 staining of the squamous cell carcinoma arising at the periphery of a KA with malignant transformation, obtained from the eyelid of a panther chameleon (**B**).

**Table 1 animals-13-00398-t001:** Signalment of five lizards that were presented with nodular skin lesions. The number and distribution of skin lesions, the interval between detection of the skin lesions and initial presentation, the follow-up period, and recurrence are specified for each case, as well as the type of ultraviolet B (UV-B) source that was provided and the average basking distance to the UV-B lighting source.

Species	Panther Chameleon 1 (*Furcifer pardalis*)	Panther Chameleon 2 (*Furcifer pardalis*)	Panther Chameleon 3 (*Furcifer pardalis*)	Veiled Chameleon (*Chamaeleo calyptratus*)	Bearded Dragon (*Pogona vitticeps*)
**Age**	5 years	4 years	4 years	3 years	6 years
**N° of nodular skin lesions at initial presentation**	18	15	3	4	2
**Location of the nodular skin lesions**	Eyelid, dorsolateral body wall	Dorsolateral body wall	Eyelid, dorsolateral body wall	Eyelid, head, dorsolateral body wall	Dorsolateral body wall
**Time between first detection and initial presentation**	2 years	6 months	3 months	2 months	4 months
**Follow-up period**	2 years	2 years	1 year	2 years	1 year
**Recurrence**	No	No	No	16	1
**UV-B source**	ExoTerra Solar Glo 80 W	ZooMed Powersun 100 W	ZooMed Powersun 100 W	Arcadia D3 Forest	JBL UV-B Spot Plus 80 W
**Average basking distance**	16 cm	15 cm	18 cm	20 cm	26 cm

**Table 2 animals-13-00398-t002:** Immunoreactive score system (IRS) [17].

A (Percentage of Positive Cells)	B (Intensity of Staining)	IRS Score (Multiplication of A and B)
0 = no positive cells	0 = no color reaction	0–1 = negative
1 = <10% positive cells	1 = mild reaction	2–3 = mild
2 = 10–50% positive cells	2 = moderate reaction	4–8 = moderate
3 = 51–80% positive cells	3 = intense reaction	9–12 = strongly positive
4 = >80% positive cells	Final IRS score (A × B): 0–12 *	

* The immunoreactive score (IRS) is calculated by multiplying the positive cells’ proportion score (0–4) and the staining intensity score (0–3).

**Table 3 animals-13-00398-t003:** Six keratoacanthomas from lizards (*n* = 6) were categorized according to the percentage of positive cells (0–4), the intensity of immunostaining (0–3), and the IRS score (0–12) (Table 2) for e-cadherin, cyclooxygenase-2 (COX-2), and pan-cytokeratin (Pan-CK).

Immunomarker	Percentage of Positive Cells					Intensity of Immunostaining				IRS Score			
	0	1	2	3	4	0	1	2	3	0–1	2–3	4–8	9–12
E-cadherin (*n* = 6)	0	0	0	0	6	0	0	0	6	0	0	0	6
COX-2 (*n* = 6)	0	0	1	4	1	0	1	4	1	0	0	6	0
Pan-CK (*n* = 6)	0	0	0	0	6	0	0	0	6	0	0	0	6

## Data Availability

The data presented in this study are available on request from the corresponding author.

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
