# Peer review of "Gross, Histologic and Immunohistochemical Characteristics of Keratoacanthomas in Lizards"

_animals, 2023, doi:10.3390/ani13030398_

Round 1

Reviewer 1 Report

The authors have carried out a case series with gross and histopathological characterization of a never described neoplasia in lizards. The paper is nicely written and contributes in building knowledge on cutaneous tumours in reptiles. I am recommending this manuscript for publication with minor revision, referred mostly to points of editorial nature, which I am listing here below:

- L58: the sentence relates SCC and chromatophoromas in squamates. However, reference #4 deals with chelonians. If authors would like to report information in squamates (discussed in this article), original reference should be preferred. (i.e., Hannon et al. 2011)

- L60: similarly, sentence deals with lizards and references concern dogs (#2) and humans (#6).

- L63: Sentence deals with dogs and references concern crocodilians (#10) and humans (#11).

- L77: It could be useful to mention analgesia provided for surgery. Indeed, the reader could have an idea of the whole protocol for clinical management.

- Figures 4, 5, 6 : stains should be mentioned in captions

- It could be interesting to add a small justification of COX2/E-cadherin/pan-cytokeratin usage and why others stains were not selected. Indeed, particularly in human medicine, a lot of immunohistochemistry protocols are described for SCC and KA.

Congratulations again for your interesting report.

Author Response

The authors have carried out a case series with the gross and histopathological characterization of a never described neoplasia in lizards. The paper is nicely written and contributes in building knowledge on cutaneous tumours in reptiles. I am recommending this manuscript for publication with minor revision, referred mostly to points of editorial nature, which I am listing here below:

- L58: the sentence relates SCC and chromatophoromas in squamates. However, reference #4 deals with chelonians. If authors would like to report information in squamates (discussed in this article), original reference should be preferred. (i.e., Hannon et al. 2011)

The remark of the reviewer is correct. Changed as suggested by the reviewer.

- L60: similarly, sentence deals with lizards, references concern dogs (#2) and humans (#6).

References number 2 and 6 (reference number 6 moved to 7 after the previous correction) concern studies carried out in reptiles and chameleons, respectively. Our apologies if we overlooked something, but the cited references seem to be correct.

  • Garner, M.M.; Hernandez-Divers, S.M.; Raymond, J.T. Reptile neoplasia: A retrospective study of case submissions to a specialty diagnostic service. Vet Clin North Am - Exot Anim Pract 2004, 7(3), 653–71.
  • Meyer, J.; Kolodziejek, J.; Häbich, A.C.; Dinhopl, N.; Richter, B. Multicentric Squamous Cell Tumors in Panther Chameleons (Furcifer pardalis). J Exot Pet Med 2019, 29, 166–72.

- L63: Sentence deals with dogs, references concern crocodilians (#10) and humans (#11).

References number 10 and 11 concern data derived from small animal medicine (dogs and cats). Our apologies if we overlooked something, but the cited references seem to be correct.

  • Abramo, F.; Pratesi, F.; Cantile, C.; Sozzi, S.; Poli, A. Survey of canine and feline follicular tumours and tumour-like lesions in 403 central Italy. J Small Anim Pract 1999, 40(10), 479–81.
  • Scott, D.W.; Miller, W.H.; Griffin, C.E. Neoplastic and Non-Neoplastic Tumors. In: Muller & Kirk’s Small Animal Dermatology, 6th ed.; Scott, D.W., Miller, W.H., Griffin, C.E., Eds.; Elsevier Health Sciences: Missouri, USA, 2001; Volume 1, pp. 1236–414.

- L77: It could be useful to mention analgesia provided for surgery. Indeed, the reader could have an idea of the whole protocol for clinical management.

We appreciate the remark of the reviewer. The analgesic protocol that was used in the lizards was added at L80 to L83:

“Anaesthesia was maintained with 1.5 - 2 % isoflurane (Isoflo®, Abbott Logistics B.V., Breda, the Netherlands) in 1 L medical oxygen with intermittent positive-pressure ventilation. Local infiltration with 2 mg/kg lidocaine SC (Xylocaine 1%, Aspen Pharma Trading Limited, Dublin, Ireland) of the incision site was performed prior to the start of the surgery, and meloxicam was administered perioperatively at 0.5 mg/kg IM every 24 hours (Metacam®, 20 mg/mL, Boehringer Ingelheim, Vetmedica GmbH, Ingelheim, Germany).”

- Figures 4, 5, 6: stains should be mentioned in captions

The stains were added to the figure captions as suggested by the reviewer.

- It could be interesting to add a small justification of COX2/E-cadherin/pan-cytokeratin usage and why others stains were not selected. Indeed, particularly in human medicine, a lot of immunohistochemistry protocols are described for SCC and KA.

The main aim of this study was to investigate general IHC characteristics of this neoplastic entity. A selection of IHC markers (KI-67, epithelial membrane antigen (EMA), CD30, pan-cytokeratin (pan-CK), E-cadherin, and cyclooxigenase2 (COX-2)) were tested based on their general use in typing SCCs and the available literature data for the use of these markers in reptiles. As none of the tissue sections and positive controls obtained from lizards were immunohistochemically positive with antisera against KI-67 and CD30 and EMA gave inconclusive results, these markers were omitted. On the other hand, immunoreaction was observed for Pan-CK, E-cadherin, and COX-2. Although several other IHC markers showed their value towards the characterization of human SCC and SCC in e.g. dogs and cats, the absence of cross-reactivity, the reptile’s skin anatomical characteristics (alpha and beta layers), or the lack of the target molecules due to evolutionary processes are common reasons why these markers are of little to no value in reptiles. Although it would be interesting to investigate the use of additional IHC markers or optimize IHC staining protocols for KAs, this was not the primary focus of this study. We have discussed the value of the IHC staining of KAs chosen in the present study in the discussion section (L341 to L350).

Congratulations again for your interesting report.

Reviewer 2 Report

Dear authors, the manuscript is good for me except some parts I recommend you to improve:

1. Delete please lines 76 - 79 in Materials and Methods. There is not any kind of analgesia (even local) so the anaesthetic protocol is not standard (delete it). Also the dosis of alfaxalone (10 mg/kg IV) is not common. It is known well that alfaxalone is used in reptiles/lizards in dosis 5 mg/kg IV.

Line 171 - delete: Recovery from anaesthesia was uneventful in all cases.

2. Table 1 - do you really think that chamaeleons/agamids are old in 3 - 6 years? They are not. Please improve the sentence on lines 266 - 267 and use ... adult males 

3. Materials and Methods and Discussion:

I did not find important details dealing with the UVB radiation - the time that lizards spent under the lamps = the lighting regime is missing! Did they spend 5 hours/days or 20 hours/day under such condition? Were the lamps controlled to be active (old lamps did not produce UV radiation at all). The UVB regime could be mentioned in the table 1 for instance.

The final part of Discussion: Lines 350 - 353 - delete it - it is only speculation not based on your data. Especially if we do no know the UV light regime and quality (age) of the lamps used.

4. Conclusions 

Delete on lines 358 - 361 Chameleons species that show ...captive environment. This conclusion is not based on your data.

Author Response

Dear authors, the manuscript is good for me except some parts I recommend you to improve:

- Delete please lines 76 - 79 in Materials and Methods. There is not any kind of analgesia (even local) so the anaesthetic protocol is not standard (delete it). Also the dose of alfaxalone (10 mg/kg IV) is not common. It is known well that alfaxalone is used in reptiles/lizards in doses of 5 mg/kg IV.

We appreciate the remark of the reviewer. The analgesic protocol that was used in the lizards was added at L80 to L83:

“Anaesthesia was maintained with 1.5 - 2 % isoflurane (Isoflo®, Abbott Logistics B.V., Breda, the Netherlands) in 1 L medical oxygen with intermittent positive-pressure ventilation. Local infiltration with 2 mg/kg lidocaine SC (Xylocaine 1%, Aspen Pharma Trading Limited, Dublin, Ireland) of the incision site was performed prior to the start of the surgery, and meloxicam was administered perioperatively at 0.5 mg/kg IM every 24 hours (Metacam®, 20 mg/mL, Boehringer Ingelheim, Vetmedica GmbH, Ingelheim, Germany).”

We agree that the IV administration of alfaxalone in saurian species is often recommended at a dose of 5 mg/kg in exotic animal formularies. We generally use 5-10mg/kg (IV) and adapt the dosing on a case-to-case basis. For longer procedures, we routinely use 10 mg/kg as it facilitates the intubation and maintenance of inhalation anaesthesia at on average lower percentages. We experience this as a highly reliable, safe, and effective dosage. In addition, several recent studies support the use of this dosage (or the limitations of lower dosages) in saurian:

  • Carpenter, J.W.; Marion, C. Exotic Animal Formulary. 5th; Elsevier Health Sciences: Missouri, USA, 2017; Volume 1, pp. 142
  • Webb, J.K.; Keller, K.A.; Chinnadurai, S.K.; Kadotani, S.; Allender, M.C.; Fries, R. Use of alfaxalone in bearded dragons (Pogona vitticeps): optimizing pharmacodynamics and evaluating cardiogenic effects via echocardiography. J Am Vet Med Assoc 2022, 261(1):126-131.
  • Perry, S.M.; Konsker, I.; Lierz, M.; Mitchell, M.A. Determining the safety of repeated electroejaculations in veiled chameleons (Chamaeleo calyptratus). J Zoo Wildl Med 2019, 50(39): 557-69
  • Scheelings, T.F.; Baker, R.T.; Hammersley, G.; Hollis, K.; Elton, I.; Holz, P. A preliminary investigation into the chemical restraint with alfaxalone of selected Australian squamate species. J Herpetol Med Surg 2011, 21(2-3):63–67.

– Delete line 71: Recovery from anaesthesia was uneventful in all cases.

Deleted as suggested by the reviewer.

- Table 1 - do you really think that chamaeleons/agamids are old in 3 - 6 years? They are not. Please improve the sentence on lines 266 - 267 and use ... adult males 

We changed the original to L275-276 as follows: “It is noteworthy that all affected lizards in the present study were adult males. Except for the bearded dragon, all chameleons had a relatively old age according to physiological lifespan reference intervals.

The bearded dragon was considered an adult animal as it can reach an average lifespan of 10-15 years. The ‘relatively old age’ refers to the chameleons based on what is generally seen in captive chameleons and is supported by the available literature:

  • Slavens F, Slavens K. Reptiles and amphibians in captivity: longevity home page, 2007. http://www .pondturtle.com/lliza.html#Chamaeleo.
  • McGeough, R. Furcifer pardalis (Panther Chameleon) – A Brief Species Description and Details on Captive Husbandry. BEMS Reports 2016, 2(2):27-38.

- I did not find important details dealing with the UVB radiation - the time that lizards spent under the lamps = the lighting regime is missing! Did they spend 5 hours/days or 20 hours/day under such condition? Were the lamps controlled to be active (old lamps did not produce UV radiation at all). The UVB regime could be mentioned in the table 1 for instance.

- Lines 350 - 353 – delete: only speculation not based on your data. Especially if we do not know the UV light regime and quality (age) of the lamps used.

We thank the reviewer for this remark. We added the following information in the text at L138 to L141 and Table 1 to address this comment.

 “All owners of the lizards included in this study replaced the ultraviolet (UV-B) lamps at regular and appropriate intervals according to recommendations. Although exact photoperiods and the specific basking behaviour could not be determined for the individual lizards, the owners stated that UV-B lighting was provided between 10 to 12 hours per day.”

- Delete on L358 - 361 Chameleons species that show ... captive environment. This conclusion is not based on your data.

Deleted as suggested by the reviewer.

Reviewer 3 Report

This is an interesting  work about a first description of keratoachantomas (KA) in three species of lizards. Biological behavior, histopathological characteristics as IHC features were described. The manuscript is well-written and features of KA well described and discussed. However, I have only some minor comments that should be addressed.

Title: I suggest to change the title in "Gross, histologic and immunohistochemical characteristics of keratoacanthomas in three lizard species".

Materials and methods:

Line 75: Did you performed any preoperative check for each animal? Did you find any abnormality? 

Line 77-80: as the Alfaxalone has not proved to be analgesic, did you included analgesic drugs in your protocol? Analgesia should be always guaranteed in any kind of surgery.

Line 110: 20 min instead of 20min.

Round 2

Reviewer 2 Report

Excellent, nice paper. Would be nice for me to use it for References and to look for this type of skin lesions in my practice.

P.S. My short comment -I am using alfaxalone for the last 15 years at least (5 mg/kg is the gold standard). But, no problem.